

# Ecological and anthropogenic factors influencing the Summer habitat use of *Bos gaurus* and its conservation threats in Chitwan National Park, Nepal

Surakshya Poudel[1], Basudev Pokhrel[1], Bijaya Neupane[2], Mahamad Sayab Miya[3], Nishan Kc[4], Chitra Rekha Basyal[5], Asmita Neupane[1] and Bijaya Dhami[6]

[1] Faculty of Forestry, Agriculture and Forestry University, Hetauda, Bagmati, Hetauda, Makwanpur, Nepal
[2] Department of Forest Sciences, Faculty of Agriculture and Forestry, University of Helsinki, Helsinki, Finland
[3] Department of Biology, Western Kentucky University, Bowling Green, KY, United States of America
[4] WWF Nepal, Baluwatar, Kathmandu, Nepal
[5] Institute of Forestry, Pokhara Campus, Tribhuvan University, Pokhara, Gandaki, Nepal
[6] Department of Biological Sciences, University of Alberta, Edmonton, Alberta, Canada

## ABSTRACT

**Background.** Gaur (*Bos gaurus*) is listed as a vulnerable species in the IUCN Red List of threatened species due to the rapid population decline caused by human pressures in their habitats. To develop successful conservation plans, it is essential to understand the relationship between Gaur and their habitats. This study aimed to investigate the factors influencing Gaur habitat use and identify and rank conservation threats in Chitwan National Park, Nepal.

**Methods.** Using line transect surveys, we recorded Gaur's presence *via* direct sightings and indirect signs (dung, footprints, horns) over an area of 176 km$^2$ in July and August 2022. We used binary logistic regression models to determine the collected ecological and anthropogenic factors influencing the occurrence of Gaur and the relative whole-site threat ranking method to rank conservation threats.

**Results.** The results revealed that the probability of Gaur occurrence increases with moderate to high canopy cover, riverine and *Shorea robusta* dominated forests, and nearest distance to road/path/firelines, while decreasing with the presence of predators. Uncontrolled fire, invasive species, human disturbances, and climate change were ranked as the most prevailing threats to Gaur in our study area. Conservation managers should implement effective habitat management interventions, such as construction of waterhole, firelines maintenance, grassland management, and control of invasive species in the potential habitats, to safeguard and maintain the sustainability of Gaur populations and associated herbivores. Future studies should consider larger geographical settings and multiple seasons, and habitat suitability assessments should be conducted to determine current and future suitable habitat refugia for Gaur and other threatened wildlife species at the landscape level.

Corresponding author
Bijaya Dhami, bdhami@ualberta.ca

## INTRODUCTION

Large-bodied species inhabiting tropical regions in developing countries face a higher risk of extinction due to anthropogenic pressure (*Schipper et al., 2008*; *Oberosler et al., 2020*; *Regmi et al., 2022*). The extensive habitat loss and degradation prevailing in these regions is one key contributory factor to this, which makes it more challenging for large mammals to meet their intricate necessities (*Ceballos & Ehrlich, 2002*; *Kinnaird et al., 2003*; *Cardillo et al., 2005*). For the long-term and immediate conservation of such species, information regarding ecological and anthropogenic factors affecting their occurrence is crucial (*Klaassen & Broekhuis, 2018*). However, such information is limited, especially for the Gaur (*Bos gaurus*), the largest living wild ungulate belonging to the family Bovidae.

Gaur, also known as Indian Bison or *Gauri Gai* in Nepal, are mega-herbivores who have important role in maintaining a healthy ecosystem (*Sankar et al., 2013*). They typically live in a herd of five to twelve individuals (*Menon, 2014*), and three different association have been observed, including solitary adult males, bull groups, and mixed herds (*Ashokkumar et al., 2011*). They are sexually dimorphic, and their sexes can be recognized precisely after the age of two years (*Ahrestani & Prins, 2011*). Young Gaur's horns are typically black, but as they mature, they turn white (*Ahrestani, 2018*). Gaur usually requires a larger habitat area and a higher amount of food than other smaller herbivores (*Chetri, 2006*). Their habitat is characterized by large and relatively undisturbed closed canopy forest areas (evergreen, semi-evergreen, and moist deciduous forests) with hilly terrain below an elevation of 1,800 m (*Nowak, 1999*), but sometimes during the dry season, they also occur in the plain area (*Choudhury, 2002*). They are predominantly grazers and browsers that generally prefer green grasses, as well as coarse, dry grasses, forbs, and leaves (*Duckworth et al., 2016*).

Gaurs are extant in Nepal, India, Bangladesh, Myanmar, Thailand, Bhutan, Cambodia, China, the Lao People's Democratic Republic, Malaysia (Peninsular Malaysia), and Vietnam (*Duckworth et al., 2016*). They are categorized as vulnerable in the IUCN Red List of threatened species and under Appendix A of CITES (*Duckworth et al., 2016*). In Nepal, they are strictly protected species under Schedule I of the National Park and Wildlife Conservation Act 1973 (*GoN, 1973*). The global population of Gaur is estimated to be 15,000–35,000 individuals (*Duckworth et al., 2016*). In Nepal, they occur in an isolated pocket of the Chure Bhabar region, Parsa National Park, Chitwan National Park (CNP), and Trijuga Valley of Nepal, with a total population of 473 individuals (*DNPWC, 2020*).

The global population of Gaur is expected to decline by 30% over the next three generations due to a wide array of conservation threats (*Duckworth et al., 2016*). Both globally and in Nepal, the conservation threats to the Gaur population include habitat loss and fragmentation, hunting and poaching for its meat and horns, epidemic diseases such as Rinderpest, human-Gaur conflict, food competition with domestic livestock, and predation of its calves primarily by leopards and tigers (*Choudhury, 2002*; *Ashokkumar et al., 2011*; *Jnawali et al., 2011*; *Duckworth et al., 2016*; *Ahrestani, 2018*). Apart from these threats, their habitat ecology is poorly studied, which impedes the evidence-based conservation of Gaur in Nepal (*DNPWC, 2020*).

To our knowledge, very few studies have been undertaken on Gaur in Nepal, focusing on multiple themes that include diet and food habits (*Chetri, 2003*; *Chetri, 2006*); population assessment at regular intervals of years by park authorities (*CNP, 2021*; *PNP, 2022*); and general ecology (*Khadka, Acharya & Chaudhary, 1997*). Aside from population status, Nepal currently lacks up-to-date research on other aspects of Gaur, particularly the ecological and anthropogenic factors that influence its occurrences. Although the Gaur Conservation Action Plan (2020–2024) for Nepal has been undertaken, robust studies are still required to fully comprehend the current state of Gaur's habitat (*DNPWC, 2020*). The contiguous habitat of Chitwan and Parsa National Parks and their adjoining areas harbor the largest population of Gaurs (*DNPWC, 2020*). A total of 388 individuals of Gaur were recorded by the CNP in 2021, of which 71 were males, 58 were females, 42 were calves, and others were unidentified individuals (*CNP, 2021*). Different ecological and anthropogenic factors such as habitat types, vegetation types, terrains, canopy cover, ground cover, distance from human settlements, water bodies, roads, aspects, and elevation can influence the Gaur's habitat selection (*Imam & Kushwaha, 2013*; *Sankar et al., 2013*). These factors may have an impact on their long-term distribution and habitat occupancy. Understanding the various factors that the species avoids or prefers will help the concerned authorities to implement science-based conservation interventions (*Manly et al., 2002*; *Nishan et al., 2023*). Therefore, the current study was carried out with the primary aim of improving Gaur conservation by gaining a better understanding of the important ecological and anthropogenic factors influencing habitat selection, as well as their associated conservation threats in the CNP of Central Nepal. The study's findings will assist park authorities and concerned conservation entities in effectively planning and executing the conservation program for Gaur, as well as minimizing the prevalent threats throughout their habitats. Furthermore, this study can be the first of its kind on the Gaur, and it will serve as a baseline for future research in other potential Gaur habitats in Nepal.

## MATERIALS & METHODS

### Study area

The study was conducted in CNP; the first national park of Nepal and one of the UNESCO World Heritage sites (Fig. 1). CNP and its buffer zone is situated in the southern part of Central Nepal which spreads over Chitwan, Nawalparasi, Parsa, and Makwanpur districts. The park (N27°20′19″ to 27°43′16″ longitude and E83°44′50″ to 84°45′03″ latitude) occupies an area of 952.6 km$^2$ in the Rapti Valley of the Siwalik physiographic region, while the buffer zone (N27°28′23″ to 27° 70′38″ longitude and E 83° 83′98″ to 84° 77′38′latitude) extends 729.37 km$^2$ area. Its altitude ranges from about 100 m in river valleys to 815 m in Siwalik hills (*Bhuju et al., 2007*).

The park is recognized as a global biodiversity hotspot, comprising healthy habitats for several globally threatened species, including the Bengal tigers (*Panthers tigris*), greater one-horned rhino (*Rhinoceros unicornis*), Asian elephants (*Elephas maximus*), gharials (*Gavialis gangeticus*), and Gaur (*CNP, 2015*). The contiguous habitat of Chitwan and Parsa National Parks and adjoining areas harbor the largest population of vulnerable fauna Gaur (*DNPWC, 2020*).
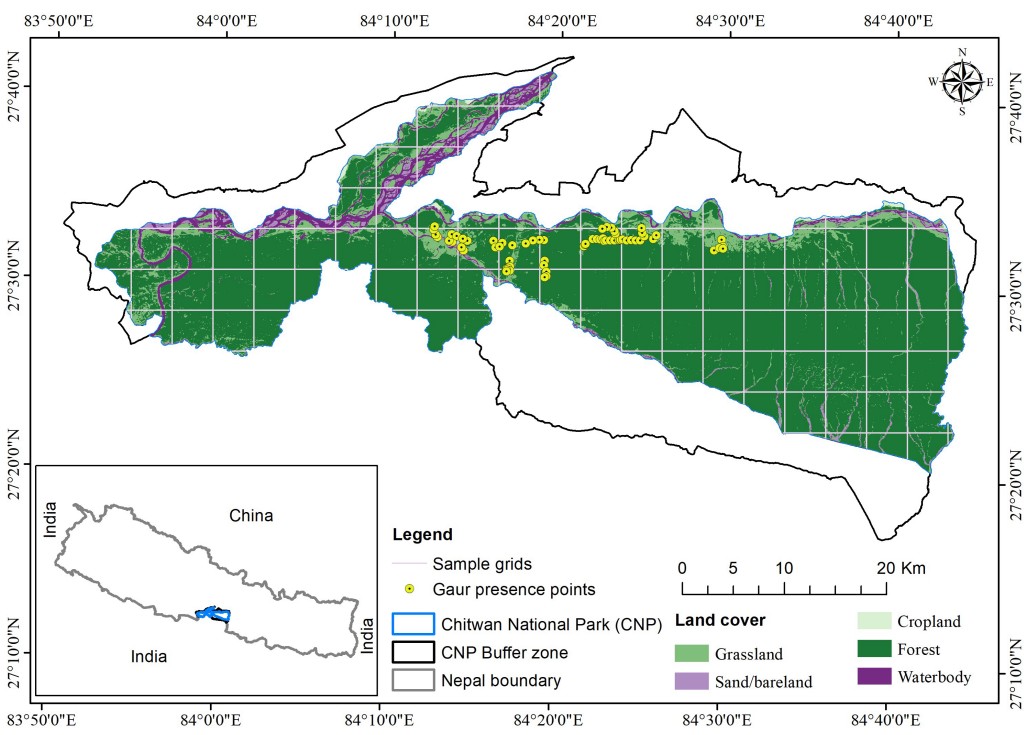

**Figure 1** **Study area map showing the Gaur presence locations in Chitwan National Park.** The land cover map was prepared by using Landsat 9 imageries (USGS).

## Data collection

### Preliminary survey

Initially, in late July 2022, the first phase of the preliminary field visit took place over five days. This involved consultation with officials of CNP, National Trust for Nature Conservation staff, Buffer Zone Management Committee member, field biologists, and nature guides to understand the overview of current distribution and identify potential sites of occurrence. Moreover, based on their insights and experiences of the study area, the research team documented the existing threats affecting the Gaur population and its habitat. Subsequently, a preliminary site visit was conducted. With guidance and support from park game scouts and nature guides, the team surveyed potential locations. We also recorded GPS points of the Gaur herd; only fresh signs (dung/scat, and footprint/pugmark) (1–5 days old) (*Kshettry, Vaidyanathan & Athreya, 2018*; *Variar et al., 2023*) of the Gaur, tiger and common leopard were noted using the knowledge and experience of field guides. The rationale for documenting signs of the common leopard is rooted in reports indicating that while adult gaurs are less likely to fall prey to leopards, instances of calves and subadults being fed upon by common leopards have been documented in India (*Kshettry, Vaidyanathan & Athreya, 2018*). Later, coordinates were plotted in ArcGIS 10.8.1 version, enabling the selection of sites to plan for a detailed field survey *i.e.,* selection of grid and transect layout.
The second phase of the study resumed in mid-August, spanning over 15 days. Its objective was to conduct transect surveys and rank conservation threats identified during the preliminary field visit and consultations with key stakeholders. We opted for the mid-August season for our detailed field survey based on the meteorological pattern in Nepal. Summer precipitation sharply rises from June, peaking in July, and begins to taper off from August onwards (*Sharma et al., 2020*).

## Detailed field survey and recording of the habitat characteristics

To access the species-habitat relationship, we adopted 'use *vs* available' resource selection functions at the population level as our sampling strategy (*Manly et al., 2002*; *Johnson et al., 2006*; *Northrup et al., 2013*). Our sampling strategy involved overlaying a square of 4km ×4km geographical grid cells (hereafter sampling units) (*Burnham, Anderson & Laake, 1980*) over the entire CNP using ArcGIS version 10.8.1 (*ESRI, 2020*). Among the 59 sampling units, we purposively selected 11 sampling units namely Kasara, Dhoba, Bankatta, Dhruba, Sukhibar, Bhimle, Dumariya, Jarneli, Bhimpur, nearby Ghatgai and nearby Dhoba. The selection of sampling units was informed by both the historical distribution patterns of Gaur in CNP, and incorporating valuable insights gleaned from stakeholder consultations during preliminary field visits. To expedite the data collection process, the selected sampling grids were partitioned into two sectors: eastern (Dumariya, Jarneli, Bhimpur, Nearby Ghatgai and Nearby Dhoba) and western (Kasara, Dhoba, Bankatta, Dhruba, Sukhibar and Bhimle). To ensure standardized data collection across sampling units and minimize temporal bias, we employed a double-team approach. Two survey teams, each consisting of one co-author and three experienced field technicians (minimum three years of experience), conducted simultaneous surveys in the eastern and western sectors.

Within each selected sampling unit, we implemented two line transects, each spanning 4 km in length (*Bayani & Watve, 2016*). These transects were positioned 200 m apart to ensure non-overlapping sampling areas. To enhance efficiency, the team of four individuals was further divided into pairs, enabling simultaneous surveying of each transect.

Each pair recorded all live sightings of the animals within a 50 m distance and indirect signs (dung, and footprints) within a 5 m distance on either side of the transect (*Pokharel & Chalise, 2010*; *Khulal et al., 2021*; *Neupane, Chhetri & Dhami, 2021*; *Neupane et al., 2022*). This paired approach minimized the potential for double-counting individuals, as both transects progressed in the same direction. GPS coordinates of direct sightings' locations and indirect signs were recorded to prepare the distribution map using Arc GIS 10.8.1 version (*ESRI, 2020*). The survey was conducted for 15 days, following the peak activity period of Gaur (*Gad, 2012*).

Upon direct sighting of a Gaur, we marked the sighting location using a prominent topographical feature, and subsequently established a 5-meter radius circle with the sighting point as the center. Likewise, for indirect signs such as dung and footprints, we employed the same methodology, creating a 5-meter radius circle with the sign at its center (*Yahnke, 2006*; *Bernard et al., 2014*). For each of those circles, one additional circle plot with the same size was set with the center localized 100 m apart from the former circle in
a randomly chosen direction (*Neupane, Chhetri & Dhami, 2021*). These plots represented average habitat samples, independent of the presence/absence of the Gaur. In each of these circular plots, the presence of Gaur was set to 1, if Gaur or any indirect sign was observed (Used plots), or to 0, if not (Habitat availability plot). In circles of both types (Used plots and Habitat availability plots), nine habitat parameters were measured namely canopy cover, ground cover, presence/absence of trees, presence/absence of predator sign, nearest distance from water body, road/path/firelines and settlement, habitat type and depth of leaf litter (see Table 1 for detail) to determine their effect on the probability of Gaur occurrence in a particular habitat. The canopy cover was measured with the help of a spherical crown densiometer. Similarly, ground cover, presence/absence of trees (any tree > 10 cm in diameter), presence/absence of predator signs (fresh scat, and pugmarks) (1–5 days old), and habitat type were determined based on visual observations by the team members possessing similar field experiences, similar to method adopted by *Dhami et al. (2023a)*; *Dhami et al. (2023b)*. The nearest distance to the water body, road/path/firelines, and settlement was calculated using the Nearest tool in Arc GIS 10.8.1 version about the presence location of the species. The shapefile of the water body was extracted using DEM (Digital Elevation Model, 12.5 m resolution) and Landsat Image 8 (*USGS, 2021*). Likewise, the shapefiles of roads/paths/firelines and settlements were extracted from an Open street map (*OSM, 2021*). The depth of leaf litter was measured following the method suggested by *Marimon-Junior & Hay (2008)*.

## Identification of conservation threats

After conducting thorough preliminary consultations with stakeholders and comprehensive field visits, supplemented by a review of pertinent documents including the Gaur Conservation Action Plan for Nepal (2020–2024), a total of eight significant conservation threats to the Gaur were identified. Subsequently, a workshop convened, gathering officials from CNP ($n = 2$), staff from the National Trust for Nature Conservation ($n = 1$), member of the Buffer Zone Management Committee ($n = 1$), field biologists ($n = 3$), and nature guides ($n = 2$). Participants were equipped with a printed list detailing the eight identified threats and were tasked with assigning each threat a rating on a scale of 1 to 10, indicating its impact on both the Gaur population and its habitat. With 10 representing the most severe threat and 1 indicating the least impact, participants were empowered to rank each threat accordingly. Following this evaluation, the average score for each threat was calculated, selecting top five threats for the subsequent relative threat ranking procedure (Annex).

## Data analysis

### Factors influencing habitat selection by Gaur

A binomial distribution model with a logit link function ($\log\left(\frac{y}{1-y}\right)$) was used to determine the ecological and anthropogenic factors influencing the presence of Gaur in the study area using R version 4.0.3 (*RCore Team, 2020*). The dependent variable was the presence/absence of Gaur and its signs at observed locations depending on the type of circle: 1–Used plot and 0–Habitat Availability plot. The predetermined habitat variables namely canopy cover, ground cover, nearest distance to a (water body, settlement, road/path/firelines), depth of

**Table 1  Detailed information on variables used in the logistic regression model.**

| Variable | Variable type | Variable category | Values | Data source |
|---|---|---|---|---|
| Presence or absence of Gaur | Dependent | Categorical | • Presence = 1<br>• Absence = 0 | Field survey |
| Ground cover % [GC] | | Categorical | • Low (0–25%) = 1<br>• Moderate (26–50%) = 2<br>• High (51–75%) = 3<br>• Dense (76–100%) = 4 | Field survey |
| Canopy cover % [CC] | | Categorical | • Low (0–25%) = 1<br>• Moderate (26–50%) = 2<br>• High (51–75%) = 3<br>• Dense (76–100%) = 4 | Field survey |
| Habitat type [HT] | | Categorical | • Grassland = 1<br>• *S. robusta* dominated forest = 2<br>• Mixed forest = 3<br>• Riverine forest = 4 | Field survey |
| Depth of leaf litter [DL] (cm) | | Continuous | Range (0.1–0.8) | |
| Nearest distance to the water source [WD] (m) | | Continuous | Range (12–622) | *USGS (2021)* |
| Nearest distance to the road/path/fire-lines [RD] (m) | | Continuous | Range (5–1500) | *OSM (2021)* |
| Nearest distance to the settlement [SD] (m) | Independent | Continuous | Range (977–6566) | *OSM (2021)* |
| Presence/absence of tree [Pat] | | Categorical | • Presence = 1<br>• Absence = 0 | Field survey |
| Presence/absence of predator sign (pugmark, scat) [Pap] | | Categorical | • Presence = 1<br>• Absence = 0 | Field survey |

leaf litter, presence/ absence of trees, presence/absence of predator signs, and habitat type were included as independent variables. For the independent variables, a multi-collinearity test was executed based on Variance Inflation Factor (VIF) analysis (*Montgomery, Peck & Vining, 1982*) taking VIF less than 10 (*Bowerman & O'connell, 1990*) using the package 'Faraway' (*Boomsma, 2014*). After confirming no multicollinearity among the selected variables, we created models for every possible subset of potential predictors using the R package "MuMIn" (*Barton & Barton, 2015*). The models were then ranked using the Akaike information criterion (AIC), adjusted for small sample sizes (AICs), as suggested by *Anderson & Burnham (2002)*. The modeled coefficients (Estimate, Std. Error, z value, and (Pr (>|z|)) were generated *via* model averaging (models with $\Delta AIC \leq 2$) for each independent variable using the package "AICcmodavg" (*Mazerolle & Mazerolle, 2017*). Likewise, the predictive precision of the best-suited model was examined using the R package "ROCR" (*Sing et al., 2005*). The values between 0.7–0.8 are considered acceptable discrimination ability, 0.8–0.9 is considered excellent, and more than 0.9 is considered superior (*Hosmer, Lemeshow & Lemeshow, 2000*).

## Analysis of conservation threats

All the conservation threats that were identified from the detailed field survey and the first phase of the preliminary survey (consultations with local stakeholders) were listed and ranked through the relative whole-site threat ranking method suggested by *WWF (2007)* by considering criteria: scope, severity and urgency (*Neupane, Chhetri & Dhami, 2021*) (see Table 2 for detail). This method was also adopted by *Chhetri et al. (2020)* and *Khulal et al. (2021)* for threat identification and ranking. Together we identified six major threats which were then assigned a relative ranking from high (5) to low (1) referring to 5 as the most severe threat and 1 as least for all three criteria. Later, the ranking scores of each criterion were summed up and were reclassified into four classes very high, high, medium, and low.

# RESULTS

## Factors influencing the occurrence of Gaur in Chitwan National Park

Initially, three component models were generated through model averaging (models with $\Delta AIC \leq 2$). Among these three component models, the model with an additive effect of canopy cover, nearest distance to road/path/firelines, habitat type, and presence or absence of predator established itself as best fitting model with the lowest AICc = 312.56 and highest AIC weight 0.53. The competing model included canopy cover, depth of leaf litter, nearest distance to road/path/firelines, habitat type, and presence or absence of predator and had an AICc value of 314.15 and AIC weight of 0.24 (Table 3). Further, the area under the Receiver Operating Curve (ROC) for the best-fit model was calculated to be 0.75 with an accuracy value of 0.79 (79%) indicating an acceptable discriminatory ability (Fig. 2).

Among the predetermined nine variables, the model-averaged coefficient revealed that canopy cover, nearest distance to road/path/firelines, habitat type, and presence or absence of predator only influenced the occurrence of Gaur significantly (Table 4). Results showed that the probability of Gaur occurrence increases with an increase in moderate to high canopy cover, riverine and *Shorea robusta* dominated forest, and nearest distance to road/path/firelines, whereas decreases with an increase in the presence of the predator.

## Major threats to the survival of Gaur

Among the five major threats identified during the field survey and preliminary survey (consultations with local stakeholders), uncontrolled burning was identified as a critical threat to the survival of Gaur. Similarly, invasive species and human disturbance were identified as major threats. Likewise, climate change and infrastructure development were ranked as medium and low threats to the species, respectively (Table 5).

# DISCUSSION

Our study provides baseline information on the ecological and anthropogenic factors influencing the distribution of Gaur and the conservation threats in the CNP of Nepal. This information has been lacking in the study area which can be very useful for protected area managers and concerned conservation authorities for formulating the management plans for conserving this least studied species and its habitats in the study area. Despite being one of the major prey species of endangered and threatened species of tiger, there

**Table 2    The Relative Whole-Site Threat Ranking method under scope, severity, and urgency criteria.**

| Criterion | Scale | Classification | Definition |
|---|---|---|---|
| Scope | 4 | Very High | The threat is expected to impact the target through all or most (71–100%) of its inhabitants. |
| | 3 | High | The threat is expected to impact the target through much (31–70%) of its inhabitants. |
| | 2 | Medium | The threat is expected to impact the target through some (11–30%) of its inhabitants. |
| | 1 | Low | The threat is expected to impact the target through a small portion (1–10%) of its inhabitants. |
| Severity | 4 | Very High | Within the scope, the threat is expected to damage or remove the target or lessen its population by 71–100% in 10 years or 3 generations. |
| | 3 | High | The threat is expected to deteriorate/lessen the target or lessen its population by 31–70% in 10 years or 3 generations. |
| | 2 | Medium | The threat is expected to moderately deteriorate/lessen the target or lessen its population in 10 years or 3 generations. |
| | 1 | Low | The threat is expected to slightly deteriorate/lessen the target or lessen its population by 1–10% in 10 years or 3 generations. |
| Urgency | 4 | Very high | The consequence of the threat cannot be driven back whose target is impossible to be replaced or would take more than 100 years to attain. |
| | 3 | High | The consequence of the threat can be driven back whose target can be restored within 21-100 years. |
| | 2 | Medium | The consequence of the threat can be driven back whose target can be restored within 6-100 years. |
| | 1 | Low | The consequence of the threat can be driven back whose target can be restored within 0-5 years. |

**Table 3    Component models (models with ΔAIC ≤ 2) describing the variable influencing the occurrence of Gaur.**

| Component models | df | Loglik | AICc | ΔAIC | Weight |
|---|---|---|---|---|---|
| Detection ~(canopy cover + nearest distance to road/path/firelines + habitat Type + presence or absence of Predator) | 9 | −146.99 | 312.56 | 0.0 | 0.53 |
| Detection ~(canopy cover + depth of Leaf litter + nearest distance to road/path/firelines + habitat Type + presence or absence of Predator) | 10 | −146.73 | 314.15 | 1.6 | 0.24 |
| Detection ~(canopy cover + nearest distance to road/path/firelines + habitat Type + presence or absence of Tree + presence or absence of Predator) | 10 | −146.73 | 314.16 | 1.6 | 0.24 |

has been less attention and studies on this large ungulate species in Nepal. The presence or absence of Gaur in a particular habitat depends on the availability of different habitat characteristics preferred by the species.

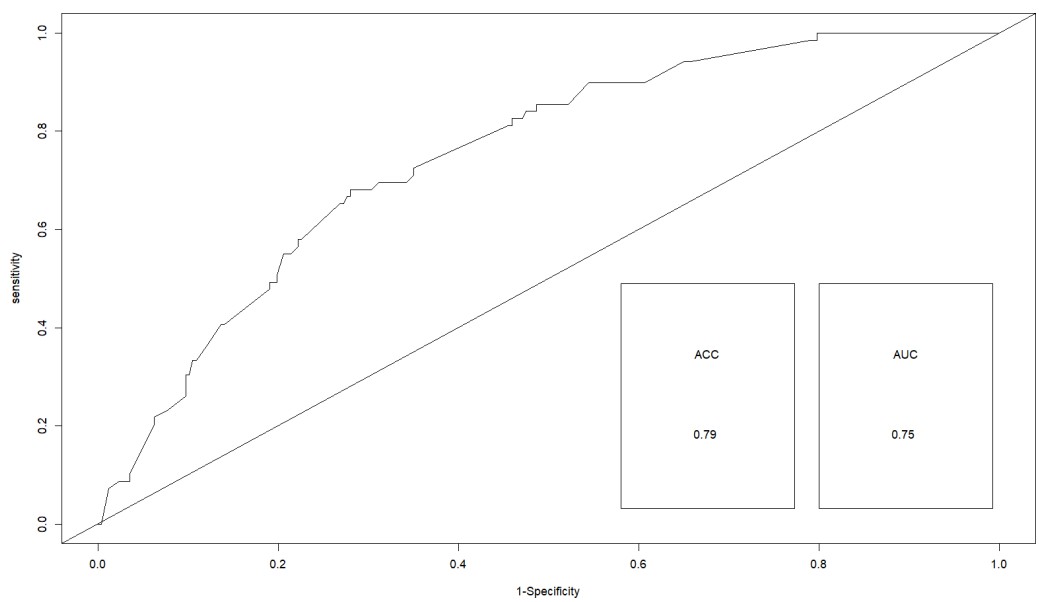

**Figure 2** **Receiver operating curve for best fitting modeling under binary logistic regression.**

**Table 4** **Model averaged coefficients describing the occurrence of Gaur in Chitwan National Park.** The variables influencing the occurrence of Gaur are denoted with a * sign.

| S. N | Predictors | Estimate ($\beta$) | Standard Error (S.E) | Z-value | Pr (>|z|) |
|---|---|---|---|---|---|
| 1 | Intercept | −2.8533445 | 0.7629528 | 3.726 | 0.000194[***] |
|  | Factor [CC]2 | 1.7092743 | 0.7123025 | 2.400 | 0.016411[*] |
| 2 | Factor [CC]3 | 1.6326545 | 0.6705469 | 2.435 | 0.014900[*] |
|  | Factor [CC]4 | 0.7372586 | 0.6926078 | 1.064 | 0.287117 |
|  | Factor [HT]3 | −0.3005559 | 0.8317088 | 0.361 | 0.717822 |
| 3 | Factor [HT]4 | 1.9618749 | 0.7577388 | 2.589 | 0.009622[**] |
|  | Factor [HT] 2 | 1.2969515 | 0.5996149 | 2.163 | 0.030543[*] |
| 4 | DL | −1.0750076 | 1.4806190 | 0.726 | 0.467806 |
| 5 | DR | 0.0011637 | 0.0003428 | 3.395 | 0.000686[***] |
| 6 | Factor [Pap]1 | −1.3561172 | 0.5853594 | 2.317 | 0.020519[*] |
| 7 | Factor [Pat]0 | −0.8052494 | 1.1900612 | 0.677 | 0.498631 |

**Notes.**
[*,**,***] $Pr (> jzj)$ represents $p$-values with the following significance codes: '***' 0.001 '**' 0.01 '*' 0.05.
Factor [CC, 2] means Moderate canopy cover; Factor [CC, 3] means High canopy cover; Factor [CC, 4] means Dense canopy cover; Factor [HT, 3] means Mixed forest; Factor [HT, 4] means Riverine forest; Factor [HT, 2] means Sal dominated forest; DL, means Depth of leaf litter; DR, means nearest distance to road (road/path/firelines); Factor [Pap, 1] means presence of Predator; Factor [Pat, 0] means absence of tree.

## Factors influencing the probability of occurrence of Gaur

Food, water and cover are the essential habitat factors that mostly influence the occurrence and distribution of Gaur (*Ebil Yusof, 2009*; *Prayoon et al., 2024*). However, the preferable habitats of Gaur are site-specific that are influenced by type of vegetation, forest cover and proximity to water bodies. Among the pre-determined nine predict variables that were chosen in our study, we found that the probability of occurrence of Gaur was

**Table 5** Threats ranked through the Relative Whole-Site Threat Ranking under four scales: very high (≥ 11.26); high (≥7.6 –≤11.25); medium (≥ 3.76 –≤ 7.5); low (≤ 3.75).

| S.N. | Threats | Scope | Severity | Urgency | Total | Category |
|------|---------|-------|----------|---------|-------|----------|
| 1 | Uncontrolled burning | 4 | 5 | 5 | 14 | Very High |
| 2 | Invasive species | 3 | 4 | 4 | 11 | High |
| 3 | Human disturbance | 5 | 3 | 2 | 10 | High |
| 4 | Climate change | 2 | 2 | 3 | 7 | Medium |
| 5 | Infrastructure development | 1 | 1 | 1 | 3 | Low |
| | Total | 15 | 15 | 15 | 45 | |

significantly influenced by four of them, where it increased with increase in moderate to high canopy cover, Riverine and *S. robusta* species dominated forest, nearest distance to road/path/firelines, while decreased with increase in presence of predator.

Our study showed that Gaur preferred moderate to high canopy cover. The finding is similar to the study of *Paliwal & Mathur (2012)*, who found the presence of Gaur showing a positive relation with canopy density classes less than 30% and 40–60% in Tadoba-Andhari Tiger Reserve (TATR), India. Unlike other smaller herbivores preferencing heterogeneous habitats, Gaurs are selective in their habitats and prefer mixed hardwood forests with the dominant species *Shorea robusta*, which has also supported the findings of (*Bhattarai & Kindlmann, 2012*). Likewise, a study on Gaur conducted in the summer season also found that it preferred grassland and *S. robusta* forest habitats in the Bandhavgarh Tiger Reserve of Central India (*Sankar et al., 2013*). Moreover, *Ariffin et al. (2021)* found Gaur more preferring riverine areas in Peninsular Malaysia. *Steinmetz (2004)* reported Gaur more commonly from mixed deciduous, dry dipterocarp, and semi-evergreen forest, in the Xe Pian National Protected Area in Lao PDR. *Zangmo et al. (2017)* reported more Gaur from subtropical warm broadleaved forests in Bhutan. Overall, deciduous forest provides a more favorable habitat for Gaur and other herbivores in Asia (*McShea, Davies & Bhumpakphan, 2011*). Our study has also identified similar results of Gaur preferring the Riverine and *S. robusta* species habitats. However, the studies conducted in different areas of India mentioned that it did not prefer dense forests (*Imam & Kushwaha, 2013*; *Sankar et al., 2013*; *Haleem & Ilyas, 2018*). Similarly, in contrast to our study, it was mentioned that Gaur preferred habitats with grasses, herbs, and shrubs and avoided woody species habitats (*Gad & Shyama, 2011*). Other studies have found that this species preferred habitat areas with low shrub and canopy cover and high grass cover (*Goswami, 2007*; *Prayoon et al., 2024*). *Paansri et al. (2022)* also noted that Gaur prefers grassland with lower canopy cover in Thap Lan National Park, Thailand. Likewise, *Zangmo et al. (2017)* found that Gaur prefers open areas in the Royal Manas National Park, Bhutan. Furthermore, *Pla-ard et al. (2022)* observed Gaur using dry evergreen forests more than other types, in Thailand. *Prayoon et al. (2021)* predicted Gaur presence more in moist evergreen forests (past), bamboo forests (present), and secondary forests (future), in Thailand. The contradictory findings of our study might be due to fewer grass species available during our survey period in the study area, which could attract the species in the forest area for feeding on barks, fallen leaves, and fruits (*Gad & Shyama, 2009*). There might be other possibilities that the

species signs might be observed in the *S. robusta* forest areas with moderate to high canopy cover because they might have come to these areas for rest after feeding the nearby grasses and forbs species.

Gaurs are sensitive to human disturbances, so they usually prefer habitats located far from human disturbances including roads, paths, and settlements (*Imam & Kushwaha, 2013*). Our study further depicted that, distance to road/path/firelines has affected most significantly in the detection of Gaur. Further nearest distance to the road in between (0–400) m was seen to have a higher presence of Gaur. It was found that road distance is an important factor contributing about 15%, where habitat suitability increases away from roads for European Bison (*Kuemmerle et al., 2018*). *Pla-ard et al. (2022)* also reported Gaur appearing more near the road than far from the road, and the distance from the road impacted the chance of occurrence at 6.6%, in Khao Yai National Park, Thailand. Another study conducted in Yellowstone National Park also reported that bison (similar species to Gaur) mostly traveled on those land areas where there was availability of streams (*Bruggeman et al., 2007*). Gaur though prefers living in the core area, as reported by *Trisurat et al. (2010)*, and travels through the road seasonally. Thus, this might be the reason that the indirect presences were observed nearest to road areas. Additionally, the roads inside the Gaur habitats are usually the fire lines and paths that are used by Gaur for travelling across different habitat patches inside the park.

In this study, the presence of predators was found to be affecting the habitat use of Gaur as the population of Gaur declined with its increase. Gaurs can quickly hear and detect the dangers of predators (*DNPWC, 2020*). So, they preferred to stay in the sites that are at lower predator risk as shown by our study. Tiger pug marks were spotted at different places in the study area. In CNP, there are 128 Tigers according to a recent survey (*DNPWC and DoFSC, 2022*). *Karanth et al. (2004)* also mentioned Gaur being preyed on by the largest carnivores of tropical forests like tiger *(Panthera tigris)* and leopard *(Panthera pardus)* in India. Tiger's main prey are ungulates, including Gaur (*Paansri et al., 2022*; *Suksavate et al., 2022*). During the diet analysis of Tiger in CNP, *Bhandari, Chalise & Pokharel (2017)* found 2% of Gaur in its diet. Similarly, *Andheria, Karanth & Kumar (2007)* found 23.87% and 9.16% Gaur in the diet of Tiger and Leopard, respectively, in Bandipur Tiger Reserve, India. Prey species such as Gaur usually avoid predator-rich areas (*Bhattarai & Kindlmann, 2013*).

## Conservation threats to Gaur

Among the five ranked threats in our study, we identified that uncontrolled burning was the most severe threat for the Gaur. Similarly, invasive species and human disturbance were recognized as the major threats to the species in our study area. Gaurs are very sensitive to habitat destruction including uncontrolled burning. For example, slash and burn shifting cultivation of hill tribes was one of the severe anthropogenic activities for threatened the Gaur population in the north-east India (*Choudhury, 2002*). Uncontrolled burning not only severely affected the wildlife species but also increased the spread and growth of invasive species in our study area (*Murphy et al., 2013*). *DNPWC (2020)* stated that controlled burning can encourage the growth of grass in the Gaur habitat. However,

sometimes, fire spreads accidentally or intentionally by local people, in an uncontrolled manner. There are historical records of disastrous forest fires in protected areas of Nepal, including CNP, causing loss of wildlife (*Sharma, 1996*; *BBC, 2012*; *GMFC, 2014*). It is argued by some scholars (*e.g.*, *Yadav et al., 2008*; *Aryal et al., 2012*; *Sadadev et al., 2021*) that fire in unsuitable timing has negative impacts on the survival of wildlife, as well as reduces the regeneration of preferred grass species and deteriorates habitat. Recently burned areas/grassland contain forage with higher quality but in low quantity (*Allred et al., 2011*), which is more used by smaller body-sized mammals than by larger body-sized, such as Gaur (*Eby et al., 2014*; *Donaldson et al., 2018*).

We found that invasive species and human disturbances are the major threats to Gaur. Invasive alien plant species (IAPS) like *Lantana camara, Mikania micrantha, Chromolaena odorata* are depleting the forest ecosystem (*DNPWC, 2020*). It was reported that in CNP, *M. micrantha* has caused a serious threat to wildlife habitat (*Murphy et al., 2013*; *Khadka, 2017*). IAPS deteriorates the habitat quality by replacing native plants (*Schirmel et al., 2016*; *Gorchov et al., 2021*). IAPS influences the abundance of wildlife, including ungulates (*Garcês et al., 2020*; *Gorchov et al., 2021*). In a recent study in the Barandabhar Corridor Forest, it was found that the abundance of ungulates decreased with increased cover of invasive species such as *C. odorata, M. micrantha,* and *P. hysterophorus* (*Adhikari et al., 2022*).

Human disturbance is another major factor in restricting the Gaur population in the wild. The human disturbances in our study area include presence of human for illegal collection of forest resources, jungle safari vehicles, jungle walk visitors, infrastructures such as machan or view tower. The threats in our study area have negative influence in the distribution and abundance of prey species of tiger including Gaur (*Bhattarai & Kindlmann, 2013*). *Gad & Shyama (2009)* also mentioned human disturbances could obstruct the movement of Gaur. Prey species like Gaur strive to avoid anthropogenic activities (*Ahrestani, 2018*; *Suksavate et al., 2022*). Gaur was reported to avoid human presences, in the study by *Paliwal & Mathur (2012)* in RATR, India. Likewise, *Bhattarai & Kindlmann (2013)* reported an abundance of prey species like Gaur, negatively associated with human disturbances in CNP. A similar finding was mentioned by *Imam & Kushwaha (2013)* in Chandoli Tiger Reserve and by *Zangmo et al. (2017)* in Royal Manas National Park. Moreover, it is stated that habitat loss due to human interference is a significant threat to the Gaur population in Asia (*Choudhury, 2002*; *Duckworth et al., 2016*).

## CONCLUSIONS

For the sustainability and conservation of biodiversity, it is vital to determine the factors that influence species distribution. Our study assessed various factors influencing the occurrence of Gaur and its major conservation threats in CNP where the highest population of this vulnerable species exists in Nepal. Canopy cover (moderate to high), Riverine, and *S. robusta* dominated forest and the nearest distance to road, path, and fire line were found to be preferred by Gaur. Similarly, uncontrolled burning, invasive species, and human disturbances were regarded as the most severe threats to the conservation of Gaur in the

park. While our research provides valuable insights on Gaur ecology, it is important to acknowledge its limitations. The study's spatial and temporal rigor was constrained by sampling in one national park and survey over a short duration. This limited scope may not fully capture the variability in habitat preferences and threats faced by Gaurs. To address this, future research should encompass broader study at landscape level including other adjoining protected areas. In addition, survey can be extended to other seasons, which would yield more robust and comprehensive findings. Such in-depth studies would ultimately contribute to more effective conservation strategies for Gaur. We further suggest habitat suitability assessment taking ecological and anthropogenic variables to determine current and future suitable habitat refugia for Gaur and other threatened wildlife species across the landscape level.

## ACKNOWLEDGEMENTS

Firstly, we would like to thank Agriculture and Forestry University, the Department of National Park and Wildlife Conservation (DNPWC), and the CNP office for providing this opportunity to carry out the research. Similarly, we would like to express our sincere gratitude to Mr. Mahesh Neupane, Conservation Officer of CNP for helping us by guiding us throughout the research period. Further, we appreciate the help of Mr. Ajit Yadav, Mr. Bed Bahadur Khadka, Ms. Hema Bhusal, Mr. Sitaram Phuyal, Ms. Sabita Thapa, Ms. Nirvika Sapkota, Ms. Anisha Parajuli, and mahouts of CNP for their continuous support and arrangements during the fieldwork. Moreover, we would like to extend our thanks to the involved field technicians during the survey period, as well as all the stakeholders involved throughout our study.

### Funding

The authors received no funding for this work.

### Competing Interests

The authors declare there are no competing interests.

### Author Contributions

- Surakshya Poudel conceived and designed the experiments, performed the experiments, analyzed the data, prepared figures and/or tables, authored or reviewed drafts of the article, and approved the final draft.
- Basudev Pokhrel conceived and designed the experiments, prepared figures and/or tables, authored or reviewed drafts of the article, and approved the final draft.
- Bijaya Neupane conceived and designed the experiments, prepared figures and/or tables, authored or reviewed drafts of the article, and approved the final draft.
- Mahamad Sayab Miya conceived and designed the experiments, prepared figures and/or tables, authored or reviewed drafts of the article, and approved the final draft.

- Nishan Kc conceived and designed the experiments, prepared figures and/or tables, authored or reviewed drafts of the article, and approved the final draft.
- Chitra Rekha Basyal conceived and designed the experiments, performed the experiments, prepared figures and/or tables, authored or reviewed drafts of the article, and approved the final draft.
- Asmita Neupane conceived and designed the experiments, performed the experiments, prepared figures and/or tables, authored or reviewed drafts of the article, and approved the final draft.
- Bijaya Dhami conceived and designed the experiments, analyzed the data, prepared figures and/or tables, authored or reviewed drafts of the article, and approved the final draft.

### Data Availability

The data is available at Zenodo: Dhami, B. (2024). Ecological and anthropogenic factors influencing the habitat use of Bos gaurus and its conservation threats in Chitwan National Park, Nepal [Data set]. Zenodo. https://doi.org/10.5281/zenodo.10552519.

### Supplemental Information

Supplemental information for this article can be found online at http://dx.doi.org/10.7717/peerj.18035#supplemental-information.

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
