# Peer review of "Ecological and anthropogenic factors influencing the Summer habitat use of Bos gaurus and its conservation threats in Chitwan National Park, Nepal"

_PeerJ, doi:10.7717/peerj.18035_

## Round 0.1 · original submission · Major Revisions

Thank you for submitting to PeerJ. Your manuscript has been reviewed by 3 experts in the field, and they have each provided detailed comments to improve your manuscript.

Key comments that will need to address in your revision are providing a detailed methodology section and evidence for the validity of your results. In your resubmission revision please address each reviewer comment and show the corresponding line in the manuscript that has been edited.

I look forward to reading your revised manuscript!

**Language Note:** The review process has identified that the English language must be improved. PeerJ can provide language editing services - please contact us at [email protected] for pricing (be sure to provide your manuscript number and title). Alternatively, you should make your own arrangements to improve the language quality and provide details in your response letter. – PeerJ Staff

Reviewer 1 ·

Basic reporting

Professional article with clear structure and flow. References need to be rechecked for standard formatting, remove duplication and irrelevant references (such as Line 473 and 475 points to same document, Line 461,569 may be irrelevant, Line 624 is incomplete).
English is clear some mistakes such as line 364 needs to be rechecked.

Experimental design

The research uses clearly mentions the hypothesis and knowledge gap on Gaur study in Chitwan and applies a standard field survey and analysis. Data collection over 15 days in July, small sample size and assessment of conservation threats based on perception limits its scope and generalization. Authors apply standard methods to validate their findings and mention the limitations in conclusions.

Authors need to further clarify the selection of 10/11? sampling units Line 149,150 and suggested to describe clearly in map.
Clarify how were variables based on visual observations were defined and measured to standardize the data collection process ?
Clarify how presence/absence of trees was measured ( did it include all tree species or only those that are palatable by Gaur, did it also include sapling and seedlings ?)

Validity of the findings

Conservation threats identify invasive species and fire as important variables impacting the Gaur, however such variables were not included in the analysis of Gaur occurrence.
Discussion on why mixed habitat have negative relation with occurrence may be useful.
Sampling was done inside the park with relatively low traffic and disturbance, how would fireline and forest path ecologically impact occurrence of Gaur need to be discussed.

Additional comments

Tables should be self descriptive, Provide more description of the tables such as in Table 4, mention about DR means distance to road, HT1 = grassland ).

Reviewer 2 ·

Basic reporting

The article is ambiguous and technically needs lots of improvement for correct text.

Experimental design

Methods used are not sufficient and are not clear.

Validity of the findings

Strong reservation on the validity of findings.

Additional comments

General Comments:
The manuscript is superficial, confusing with questionable data quality and poor research design.
The methodology is considered insufficiently explained and the results are considered unclear and potentially misleading.
Specific comments
Line 54-57: How does Gaur play important roles in ecosystem functioning and services by promoting soil sustainability, successional processes, and plant growth, explain.
Line 58: lone males are not in the group so better to use the correct term than groups
Line 59-60: “…..their sexes can be recognized 60 after the age of two years (Ahrestani & Prins, 2011)”…..but gaurs exhibit substantial sexual dimorphism even in age below two. Justify the argument
Line 65-67: The habitat explained in the introduction section is different from the habitat of Gaur in Nepal.
Line 75: Is the population of gaur in Nepal 473 is this the estimated or censused number?
Line 92-95: How are the individual counted?
Line 126: Is it endangered?
Line 160-162: The study was carried out in July, the season with long and dense grass, and also enough rain. How do authors explore indirect signs of prey and predators in long grasses and rain? What are the methods taken to mitigate overlooking the sign in long and dense grasses and also the possibility of signs being swept by rain?
Is it possible to do a 4 km length line transect in the forest of Chitwan with a width of 10 meters this season (July) in the forest of CNP?
Line 188-189: “The major conservation threats that are hindering the survival of the Gaur were assessed throughout the detailed field survey” What does the details field survey mean? Explain clearly
The methodology for the identification of conservation threats is not clear and justifiable
Line 250: ‘S. robusta’ should be italicized and with a full genus name. What is the total number of signs of Gaur and predators that were recorded? Which species were taken into account for predators? This made the result and discussion very complex and confusing.
Line 257: “Among the five major threats identified during the field survey….” Need a clear explanation of the methodology of how field survey explores the threat it is a natural habitat and the individual population is very good as mentioned in the introduction.
326-327: “In this study, the presence of predators was found to be affecting the habitat use of Gaur as the population of Gaur declined with its increase……” Does it assume all signs of predator and gaur were of the same time? Are there temporal differences? If they have temporal variation the conclusion is not correct.
Line 357: What are the human disturbances inside the protected area especially in July when the study was taken? Generally, PA is closed for visitors from June to September. What are anthropogenic threats? Is it solid waste, noise, or other but the study area is inside National Park and prime habitat of Rhino and Tiger. This leads the result misleading or misinterpretation.
Overall, the manuscript requires significant revisions and is not qualified for publication in the current version.

.

Reviewer 3 ·

Basic reporting

The proposed paper attempts to provide baseline information and factors affecting the distribution of Gaur (Bos gaurus) in Nepal. Megaherbivores are known to be of conservation importance because they are long-ranging species and are spatial and temporal resource-dependent. Therefore, various anthropogenic factors and variations in spatial ecological requirements may impact conservation. Hence, the study would provide baseline information that may be significant for planning the conservation of the species. Thus, the paper may be accepted for publication after modification as suggested below:

Title:
Line no. 1: The study was conducted only during summer. Therefore, the authors are requested to be specific in the title and mention "summer" in the title.

Abstract:
Line no. 33 to 37: Most parameters mentioned are generic, and authors are requested to confine based on the paper's findings. The authors need to mention specific tasks to be undertaken by managers for habitat management intervention and sustainability of the Gaur population. Associated herbivores require different habitat management; therefore, what management intervention undertaken may not be suitable for these, and this should be deleted.

Introduction:
Line 89 and 103: The study has not undertaken environmental parameters; therefore, it should be replaced with "ecological."

Experimental design

Materials & Methods:
Line 110:
Authors need to provide the major forest types (LULC maps), vegetation types (NDVI maps), and other associated plant species composition types as mentioned in the papers for the study area. These layers can also be included as independent variables to refine the model(s).
Line 136: Authors need to mention the period when Phase I (preliminary survey) and II (detailed field survey) studies were undertaken, and the total days spent collecting the field data for both phases. What were the ecological attributes noted during the preliminary field visit? The authors have yet to record the activity pattern of the Gaur; therefore, it needs to be rephrased.

Line 148 to 151: Explain the basis for selecting the 11 sampling units out of 73, ensuring transparency regarding potential bias. It reflects that you have chosen 176 sq. km out of the 1000 sq. km area. Were the grids distributed throughout the area, centre, or any other place? Please mention.
[Using previous sites for sampling may have introduced bias to the study due to potential changes in resource availability over time or evolutionary adaptations in response to interspecies competition.]

Line 149: The distance between two transects is only 200 meters; therefore, double counting the same Gaur on these transects is possible. Authors are requested to provide visibility on the transects and data on how far the animals have moved away from the transect when sighted. Otherwise, authors should use data only for one transect of each sampling unit.
Line 155 to 168: In the "use and availability resource selection function", ecological attributes are recorded on use and at random. Direct and indirect sightings have been recorded at 50 m & 5 m from a distance, but where the circular plots were laid must be clarified. Authors are requested to rephrase and explain the methodology.

Line 160: The Authors need to provide the study period clearly as it appears to be a pilot study in nature conducted over a span of 15 days.

Line 170 to 173. The authors need to clarify and provide details on how ecological parameters were recorded, such as canopy cover, the definition of tree used, whether the presence/absence of trees was only recorded or some quantification was done, habitat types, depth of litter, and ground cover, etc.

Line no. 187: Authors need to provide details of what attributes were considered for identifying major
conservation threats and how these were quantified. As the study was undertaken during summer, whether authors have recorded these threats only for summer or overall. Need clarifications.

Validity of the findings

Results:

Line 233 to 248: Most of the herbivores respond due to the presence of carnivores within the surrounding habitats, and it is not clear whether authors have recorded the fresh presence/absence of carnivore signs and Gaur, so it may be relevant. Otherwise, the old carnivore signs may not be used to assess the likely impact on Gaur use. Provide clarification. The statement is repeated at 235 and 247 and requires rephrasing.
Line no. 257: Authors must provide quantified data on these threats, as they vary spatially and temporally. Otherwise, bring it under discussion instead of a separate heading.

Discussion:
Line no. 277: The occurrence of species is more related distribution of food, water, and cover resources; however, authors should emphasize this aspect, and then they should bring the factors that might influence occurrence.
Line 298-299: The authors need to rephrase this; they have yet to quantify any food resources, such as grass, herbs, and shrubs, which are not comparable.
Line 314: We need to clarify the type of road considered, as several studies have reported minimal impact of forest roads.
Line 339: Authors need to provide data for ranking these threats; otherwise, mention the likely threats under discussion.

Additional comments

Considering the value of ecological information for planning Gaur conservation, the paper may be accepted for publication after revision.

---

## Round 0.2 · Major Revisions

There are still quite a few revisions requested by Reviewer 1. Please revise your manuscript to address these requests.

Reviewer 1 ·

Basic reporting

Irrelevant literature is cited (e.g., Crowther et al., 2019), and papers are wrongly interpreted to justify their statements (e.g., Ibanez-Alvarez et al., 2022). Authors try to justify statements based on very old literature (Hobbs 1996; Shaller, 1967). They refer to the same literature repeatedly to justify different statements instead of providing a synthesis. References include articles that are not cited in the text. Blunders still exist despite clear indications to recheck the references ( e.g., Having DNPWC 2020a and DNPWC 2020b, CNP 2021 a 2021b in reference, which refer to the same documents). Despite all these, the authors respond that references have been rechecked and verified, which is very unprofessional.

Experimental design

The research questions are well-defined and within the journal's scope. However, the study is limited in its spatial and temporal rigor due to sampling in a small area and a very short duration. Limitations of the study and its impact on results should be well discussed.

Validity of the findings

The purposeful selection of 11 sites out of 59 sampling units may have potentially biased the findings. Threat identification based on the perception of limited individuals despite opportunities for primary measurements in the field raises questions about the overall statistical robustness of the findings.

Additional comments

Introduction:
Line 48: It can be agreed that large-bodied animals, in general, are more vulnerable to extinction; authors need to justify with strong reference that large-bodied species from lowlands are more vulnerable compared to similar animals from highlands and large-bodied species from developing countries are more vulnerable to extinction compared to same species from a developed country.
Line 57-62: Ungulates play an important role in ecosystem functioning. However, the evidence for Gaur's role in nutrient cycling, soil aeration, and habitat succession is not very strong and irrelevant to the current paper. The authors may want to provide a stronger context and justification for the study by rephrasing the beginning sections of the introduction or rewriting the whole introduction with relevant and latest references.
Sampling: Purposive selection of 11 sites from 59 may have biased the results. Did the data collection occur along the straight 4km transects or the 4km walking route?
Line 214: Identifying threats using the perception ranking of 9 individuals does not strengthen the validity and robustness of the findings.
The authors discuss that shorea robusta habitat and fireline/path was preferred. what number of your sampling locations included Shorea forest and firelines? Is it really preferred or is it because most of the sampling points included Shorea forest and the firelines?
Figure 1: Gaur mentions preferring riverine and Shorea-dominated forests, but the figure fails to indicate their spatial location on the map.

Reviewer 3 ·

Basic reporting

Necessary edits have been made in the revised manuscript submitted by the authors. All the comments have been addressed adequately

Experimental design

Appropriate and the comments raised have been addressed

Validity of the findings

Statistically sound and details have been adequately addressed

---

## Round 0.3 · accepted · Accept

Thank you for your revisions. The reviewer is now satisfied with your manuscript.

Reviewer 1 ·

Basic reporting

Meets the basic reporting requirements .

Experimental design

Original research with methods and limitations mentioned clearly.

Validity of the findings

Methods are clearly mentioned and data provided.

Additional comments

Although the research is limited in sampling rigor, the article is well-framed and provides updated insights on Gaur habitat use in Chitwan. Congratulations and best wishes to all the authors involved.